# Early Adverse Event Derived Biomarkers in Predicting Clinical Outcomes in Patients with Advanced Non-Small Cell Lung Cancer Treated with Immunotherapy

**DOI:** 10.3390/cancers15092521

**Published:** 2023-04-28

**Authors:** Dung-Tsa Chen, Andreas N. Saltos, Trevor Rose, Zachary J. Thompson, Ram Thapa, Alberto Chiappori, Jhanelle E. Gray

**Affiliations:** 1Department of Biostatistics and Bioinformatics, H. Lee Moffitt Cancer Center & Research Institute, Tampa, FL 33612, USA; zachary.thompson@moffitt.org (Z.J.T.); ram.thapa@moffitt.org (R.T.); 2Department of Thoracic Oncology, H. Lee Moffitt Cancer Center & Research Institute, Tampa, FL 33612, USA; andreas.saltos@moffitt.org (A.N.S.); alberto.chiappori@moffitt.org (A.C.); jhanelle.gray@moffitt.org (J.E.G.); 3Department of Radiology, H. Lee Moffitt Cancer Center & Research Institute, Tampa, FL 33612, USA; trevor.rose@moffitt.org

**Keywords:** adverse event, predictive biomarker

## Abstract

**Simple Summary:**

This paper presents a paradigm-shifted approach for adverse event (AE) analysis from classic descriptive summary into modern informative statistics to fulfill precision medicine. It defines early AE and derives a series of innovative AE biomarkers to assess grade, treatment relatedness, occurrence, frequency, and duration. Comprehensive data analysis generates opportunities for global discovery of predictive early AE-derived biomarkers. A proof of concept in two lung cancer studies demonstrates the potential clinical utility of early AE-derived biomarkers in patients with advanced non-small cell lung cancer treated with immunotherapy. The methodology also provides a useful tool to help discover predictive AE biomarkers for treatment response and survival outcomes.

**Abstract:**

Rationale: Adverse events (AEs) have been shown to have clinical associations, in addition to patient safety assessments of drugs of interest. However, due to their complex content and associated data structure, AE evaluation has been restricted to descriptive statistics and small AE subset for efficacy analysis, limiting the opportunity for global discovery. This study takes a unique approach to utilize AE-associated parameters to derive a set of innovative AE metrics. Comprehensive analysis of the AE-derived biomarkers enhances the chance of discovering new predictive AE biomarkers of clinical outcomes. Methods. We utilized a set of AE-associated parameters (grade, treatment relatedness, occurrence, frequency, and duration) to derive 24 AE biomarkers. We further innovatively defined early AE biomarkers by landmark analysis at an early time point to assess the predictive value. Statistical methods included the Cox proportional hazards model for progression-free survival (PFS) and overall survival (OS), two-sample t-test for mean difference of AE frequency and duration between disease control (DC: complete response (CR) + partial response (PR) + stable disease (SD)) versus progressive disease (PD), and Pearson correlation analysis for relationship of AE frequency and duration versus treatment duration. Two study cohorts (Cohort A: vorinostat + pembrolizumab, and B: Taminadenant) from two immunotherapy trials in late-stage non-small cell lung cancer were used to test the potential predictiveness of AE-derived biomarkers. Data from over 800 AEs were collected per standard operating procedure in a clinical trial using the Common Terminology Criteria for Adverse Events v5 (CTCAE). Clinical outcomes for statistical analysis included PFS, OS, and DC. Results: An early AE was defined as event occurrence at or prior to day 30 from initial treatment date. The early AEs were then used to calculate the 24 early AE biomarkers to assess overall AE, each toxicity category, and each individual AE. These early AE-derived biomarkers were evaluated for global discovery of clinical association. Both cohorts showed that early AE biomarkers were associated with clinical outcomes. Patients previously experienced with low-grade AEs (including treatment related AEs (TrAE)) had improved PFS, OS, and were associated with DC. The significant early AEs included low-grade TrAE in overall AE, endocrine disorders, hypothyroidism (pembrolizumab’s immune-related adverse event (irAE)), and platelet count decreased (vorinostat related TrAE) for Cohort A and low-grade AE in overall AE, gastrointestinal disorders, and nausea for Cohort B. In contrast, patients with early development of high-grade AEs tended to have poorer PFS, OS, and correlated with PD. The associated early AEs included high-grade TrAE in overall AE, gastrointestinal disorders with two members, diarrhea and vomiting, for Cohort A and high-grade AE in overall AE, three toxicity categories, and five related individual AEs for Cohort B. One low-grade TrAE, alanine aminotransferase increased (vorinostat + pembrolizumab related), was an irAE and correlated with worse OS in Cohort A. Conclusions: The study demonstrated the potential clinical utility of early AE-derived biomarkers in predicting positive and negative clinical outcomes. It could be TrAEs or combination of TrAEs and nonTrAEs from overall AEs, toxicity category AEs, to individual AEs with low-grade event leaning to encouraging effect and high-grade event to undesirable impact. Moreover, the methodology of the AE-derived biomarkers could change current AE analysis practice from a descriptive summary into modern informative statistics. It modernizes AE data analysis by helping clinicians discover novel AE biomarkers to predict clinical outcomes and facilitate the generation of vast clinically meaningful research hypotheses in a new AE content to fulfill the demands of precision medicine.

## 1. Introduction

### 1.1. Background of Adverse Event (AE)

AEs are a critical component of clinical trials for safety data to protect patients from unnecessary risk and to develop a safety profile of the drug for benefit-risk assessment. The National Cancer Institute (NCI) defines an AE as any unfavorable symptom, sign, or disease (including an abnormal laboratory finding) temporally associated with the use of a medical treatment or procedure that may or may NOT be considered related to the medical treatment or procedure. NCI has also developed the Common Terminology Criteria for Adverse Events (CTCAE) to standardize toxicity characterization. The CTCAE version 5 lists 26 toxicity categories to cover a total of 837 AE terms. Each AE is recorded from onset time to resolved time with indication of severity defined as grade 1 for mild, grade 2 for moderate, grade 3 for severe, grade 4 for life-threatening, and grade 5 for fatality, as well as status of treatment related cause (definite, probable, possible, unlikely, and unrelated).

### 1.2. Potential Clinical Utility

While the primary purpose of an AE is to assess patient safety, various studies [1,2,3,4,5,6,7,8,9,10,11,12,13,14] showed the association of AEs with clinical outcomes. For example, hypertension was associated with improved survival outcomes in advanced non-small cell lung cancer (NSCLC) patients treated with bevacizumab in combination with carboplatin and paclitaxel [8] (immunotherapy+ chemotherapy) and in patients with metastatic renal cell carcinoma treated with sunitinib [9] (targeted therapy). A high neutrophil-to-lymphocyte ratio (NLR) was correlated with worse survival in multiple immune checkpoint inhibitors (ICI)-treated malignancies [10,11,12,13]. Immune-related adverse event (irAE) was also shown to be associated with improved overall survival in melanoma patients treated with nivolumab [5]. For NSCLC, a recent study [4] reviewing over ten ICI studies suggested irAEs as potential surrogates of a better ICI efficacy.

### 1.3. Challenges in AE Analysis

#### 1.3.1. Complexity of AE Data

Standard AE reporting (e.g., the worst grade method) has been criticized as sub-optimal and inefficient to properly assess patient safety profiles [15,16,17,18,19,20,21]. Figure 1 with two cases (Patient A and B) illustrates the complexity of AE data and the limitation of the standard AE reporting. Patient A had six unique AEs with a repeated event in weight loss from grade 1 advanced to grade 2. Among them, three AEs were treatment related (weight loss, dysgeusia, and anorexia) with grade 1 developed in the first two AEs over 50 days. The worst grade method yielded only two grade 2 AEs and four grade 1 AEs. This result fails to capture (i) duration (e.g., same grade with different duration): while vomiting and nausea were listed as grade 1, both AEs occurred temporally (one day). In contrast, the duration of grade 1 dysgeusia was 64 days; (ii) recurrence: weight loss was a repeated event; (iii) change of grade and treatment relatedness: initial weight loss was grade 1 and associated with treatment, but then evolved to grade 2 and was not treatment related. Patient B had four unique AEs: fatigue, creatinine increased, dyspnea, and diarrhea with fatigue as a repeated event from a non-treatment related grade 1 (57 days) advanced to treatment related grade 2 (169 days). The worst grade method failed to address the same issues.

#### 1.3.2. Limitations of Existing Strategies

Different approaches have been used to improve the characterization of AE profiles by incorporating multiple dimensions of AE, such as the toxicity over time (ToxT) approach [21], Q-TWiST approach [22] to model mean survival time, toxicity index (TI) using a total rank metric to summarize grade frequency [23], or generalized log-rank test for recurrent events [24,25]. However, their broad application was hindered due to either a restriction to single AE analysis, sub-optimization of AE parameters, difficulty of interpretation, or uncertainty of clinical relevance [26]. The ToxT approach [21] incorporates the dimension of time into AE assessment to depict toxic effects. It includes the repeated measures method to compare the mean of AE grade (treated as a continuous dependent variable) between treatment groups over all drug cycles (each drug cycle as a fixed time point) while adjusting for within-patient variation. The Kaplan-Meier method is used to analyze AE onset for time-to-event data (e.g., time to first occurrence or worst grade). Various graphics (e.g., stream plot, butterfly plot, and event chart) are also utilized to visualize AE development over time. Potential issues [26] including (i) the proportion of AE grade could be more clinically relevant compared to mean of AE grade, (ii) drug cycle as a fixed time point could be problematic when duration of a drug cycle has large variation among patients, and (iii) time-to-event analysis for AE onset could be biased because AE could occur only to a few patients (spareness) causing the majority of patients without an AE event to be excluded in the analysis. The Q-TWiST approach [22] analyzes quality adjusted time without symptoms or toxicity. It decomposes survival time into three health states: time with toxicity (TOX), time without toxicity, relapse, or progression (TwiST), and time after tumor progression or relapse until death (REL), and then uses a weighted average of the three health states to form the Q-TwiST score. The approach takes the entire AE course to derive the overall metric and has been used for treatment comparison in multiple studies [27,28,29,30,31]. Concerns include a dependency of utility weights which could influence results, the lack of estimation at a specific time point which could diminish predictive value, and the inability of individual AE analysis. The TI approach [23] uses a total ordering to fully rank the patients based on frequency of each grade experienced. While it is a better summary index over the worst grade method, AE duration is not incorporated in the metric. The generalized log-rank test [24] uses mean frequency function to compare recurrent AE events between treatment arms, and an extended version, multivariate generalized log-rank test, for comparison of the overall AE profile. While the strength is in taking account of recurrence events by cumulative incidence rate, the duration of the AE is left behind, as well as interpretation concern [26].

#### 1.3.3. Potential Biased Analysis with Less Predictive Value

The evaluation of AE association with clinical outcome could easily lead to biased results if it is not properly analyzed. When a patient continues treatment, the chance of experiencing AE events increases. Meanwhile, patients tend to have longer PFS, OS, and treatment response if treatment is continued. On the other hand, patients with treatment discontinuation at early time will likely experience disease progression or death, and therefore reduce AE experiences. This phenomenon is called guarantee-time bias [32]. That is, the negative outcome comes first (e.g., progression or death) prior to the occurrence of an AE event. As a result, patients without an AE event will be biasedly associated with poorer outcomes. The analysis for the full spectrum of AE data (from the beginning to the end of the study) often uses the extended Cox model with time-varying covariates to address the issue [3]. While the model allows for the comparison of patients with AE events versus patients who had not yet become or never experienced an AE, grade and treatment relatedness have not yet been incorporated in the model. In addition, the evaluation of the entire duration of AE has less predictive value. Another common approach is landmark analysis at an early time point, which poses a higher predictive value. The application has shown early AE associated with clinical outcomes [4,8,10,12,13,14,33,34], such as early irAE (2–6 weeks since initial treatment) in predicting responses to nivolumab in patients with NSCLC [14] and early AE (42 days after commencement of treatment) associated with improved survival in breast cancer patients treated with lapatinib plus capecitabine [33]. However, their evaluations were limited to a small AE subset, such as irAE [14], or did not take into account of frequency, duration, or grade [14,33].

By taking these considerations with the hypothesis of AE’s clinically predictive value, an integrative approach was developed by utilizing AE-associated parameters and deriving a set of innovative AE biomarkers. Global discovery analysis showed AEs as potential biomarkers in predicting clinical outcomes.

## 2. Methods

Key strategies included the derivation of AE biomarkers, global discovery in all AE data types, early AE as predictive biomarkers, and relevant clinical outcome variables for evaluation. Two study cohorts were used to demonstrate clinical and methodologic utilities.

### 2.1. Derivation of AE Biomarkers

Three key AE elements were used to enrich AE data for biomarker development: treatment relatedness, grade, and measurement. Accordingly, 24 AE-derived biomarkers were formed to evaluate their clinical association (Table 1) based on two types of treatment relatedness, three types of grade, and four types of measurement as described below.

#### 2.1.1. Treatment Relatedness

AE is one key component to investigate experimental drug agent for safety and efficacy. CTCAE v5 uses five categories (definite, probable, possible, unlikely, and unrelated) to dictate treatment relatedness to toxicity. Common AE reports are often limited to treatment-related AE (TrAE) which covers three categories (definite, probable, and possible). Since nonTrAE (unlikely and unrelated) may be also associated with clinical outcomes, the combination of all the five categories (i.e., TrAE + nonTrAE) is often used (AE is used after mentioned to distinguish from TrAE). Thus, both AE types were used for the study: TrAE and AE.

#### 2.1.2. Grade

CTCAE v5 defines AE severity in five grade levels (1–5) from mild to fatality. Common practice classifies AE severity into low-grade (grade 1 or 2) and high-grade (grade 3 or higher). The use of any-grade (covering all grades) is also often used. Thus, these three grade types were used for evaluation: any-grade, low-grade, and high-grade.

#### 2.1.3. Measurement

AE measurement could be formed in many ways. Here, we considered a set of relevant components for an AE: (i) occurrence to indicate if an AE occurs, (ii) recurrence to indicate if an AE repeatedly occurs, and (iii) duration to indicate if an AE event is temporary or long-term. The inclusion of recurrence and duration is to enrich AE information after an AE occurrence to see if a higher recurrence or longer duration could strengthen the clinical association. Accordingly, for a set of AEs, it forms four informative metrics to characterize the AEs: (a) occurrence of any AE (yes/no): yes for any AE event and no for no AE event at all, (b) frequency of all unique AEs: sum of unique AEs that occurs, (c) frequency of all AEs (including recurrence): sum of all AEs that occurs even they are repeated, and (d) sum of all AE duration (including recurrence). The incorporation of the four metrics could enhance AE contents and help tune up the impact on clinical outcomes in different aspects from occurrence, recurrence, to duration, since they have been reported for their clinical association [3,14,24,35,36].

### 2.2. AE Data Type

#### 2.2.1. Overall AE Level

All the 837 AE terms (CTCAE v5) in a patient were summarized into the 24 AE-derived biomarkers in Table 1. For example, patient A in Figure 1 had occurrence (coded as 1) in any-grade, treatment related any-grade, low-grade, and treatment related low-grade, but no occurrence (coded as 0) in high-grade and treatment related high-grade. For the sum of all unique AEs, the patient had six events in any-grade and low-grade, three events in treatment related any-grade and low-grade, and zero events in high-grade and treatment related high-grade. For the sum of all AEs, since the patient had weight-loss recurrence (twice), it became seven events in any-grade and low-grade while the others were unchanged (same to sum of unique AE metric). The sum of all AEs duration were 451 days in any-grade and low-grade, 344 days in treatment related any-grade or low-grade. Appendix A summarized the results. Patient B in Figure 1 had occurrence (coded as 1) in any-grade, treatment related any-grade, low-grade, treatment related low-grade, and high-grade, but no occurrence (coded as 0) in treatment related high-grade. For the sum of all unique AEs, the patient had four events in any-grade and three events in low-grade, one event in high grade, one event in treatment related any-grade and low-grade, and zero events in treatment related high-grade. For the sum of all AEs, since the patient had fatigue recurrence (twice), it became five events in any-grade and four events in low-grade while the others were unchanged (same to the sum of unique AE metric). The sum of all AE duration were 572 days in any-grade, 568 days in low-grade, four days in high-grade, and 169 days in treatment-related any grade or low-grade. Appendix A summarized the result.

#### 2.2.2. Toxicity Category Level

The AE-derived biomarkers in Table 1 were also constructed in each toxicity category. While there are a total of 26 toxicity groups in CTCAE v5, distribution of AE in each group is uneven. Seven toxicity groups cover less than ten AEs (e.g., ‘Congenital, familial and genetic disorders’ has one AE and ‘Immune system disorders’ has six AEs). In contrast, five toxicity groups have more than 60 AEs for each group (e.g., ‘Nervous system disorders’ has 68 AEs and ‘Gastrointestinal disorders’ includes 122 AEs). Analysis focused on toxicity groups with large AEs (similar to the strategy for the overall AE level) because toxicity groups with few AEs could have issues pertaining to sparseness. For the example in Figure 1, patient A had AE experiences under four toxicity categories: gastrointestinal disorders (abdominal pain, nausea, and vomiting), metabolism and nutrition disorders (anorexia), nervous system disorders (dysgeusia), and investigations (weight loss). The gastrointestinal disorders occurred in any-grade and low-grade with three events for sum of all unique AEs and sum of all AEs, and a total duration of 31 days. Appendix A summarizes the result. The other three toxicity categories had one AE in each toxicity category. Thus, their AE-derived biomarkers were the same as their corresponding AE (see individual AE level). 

#### 2.2.3. Individual AE level

The AE-derived biomarkers in Table 1 were applicable to each AE. For example (Appendix A), weight-loss in patient A of Figure 1 occurred in any-grade and low-grade with one unique event, a total of two events (recurrence), and a total duration of 168 days. Vomiting, nausea, dysgeusia, abdominal pain, and anorexia occurred in any-grade and low-grade with one unique event without recurrence, and a total duration of 1, 1, 64, 29, and 188 days, respectively. Anorexia also occurred in treatment related any-grade and low-grade. One caveat is that each AE may only happen to a few patients and pose a significant sparseness challenge. Instead, once AE-derived biomarkers are discovered in overall AE level and/or toxicity category level, the associated individual AEs could be then evaluated for clinical association.

### 2.3. Early AE

Early AE has been shown to have an association with clinical outcomes [4,14,33,34]. Thus, early AE-derived biomarkers could have potential in predicting treatment response or survival outcomes and provide informative evidence for clinicians to determine the continuation of treatment to improve clinical outcome or consideration of alternative treatment strategy due to ineffective drug. Early AE was defined based on duration on-treatment. We considered early AE as an event which occurred between the initial treatment date and day 30 on-treatment, close to the length for evaluation of dose limited toxicity (DLT). In other words, any AE that occurred at or before day 30 after the first treatment was counted as an early AE. The early AE-derived biomarkers were defined accordingly using Table 1. In Figure 1, for example, patient A had no early AE event since all AEs occurred after day 30 (Appendix A). Therefore, all the 24 AE biomarkers would have a measurement score of 0. In contrast, patient B had two early AEs: fatigue and diarrhea (Appendix A and Appendix A).

### 2.4. Clinical Outcomes and Statistical Analysis

Four clinical outcomes were used to evaluate AE-derived biomarkers: progression-free survival (PFS), overall survival (OS), disease control (DC: CR + PR + SD where CR: complete response, PR: partial response, SD: stable disease, and PD: progressive disease), and duration of treatment (DOT) since most outcomes were reported to be associated with AE [1,2,3,4,5,6,7,8,9,10,11,12,13,14]. The two-sample t-test was used to test the mean difference of each AE-derived biomarker between DC and PD. Correlation analysis was used for DOT while univariate Cox proportional hazards model was used for PFS and OS to analyze each AE-derived biomarker.

### 2.5. Global Discovery Analysis

For each AE (overall AE, a toxicity category, or individual AE), each of the 24 AE-derived biomarkers was evaluated in each clinical outcome with Cox model for PFS and OS, t-test for mean difference between treatment response groups (e.g., DC versus PD), and correlation analysis for DOT (e.g., Figure 2). Analysis results of the 24 AE-derived biomarkers were then summarized by a summary plot with effect size (e.g., Hazard ratio (HR)) and p value to indicate magnitude of clinical association (e.g., Figure 3). The p value was further utilized to identify significant AE biomarkers (*p* < 0.05) to determine a clinical association for the corresponding AE in summary of treatment relatedness and grade level. Given the massive analysis results (>70,000 results: 24 biomarkers × 4 outcomes × 800 AEs), the strategy efficiently generated summary reports for easy interpretation (Appendix A). For example, Appendix A listed low-grade and any-grade overall AE (TrAE) associated with improved PFS and OS and high-grade overall AE associated with poorer PFS and OS (including TrAE) in one study cohort.

### 2.6. Study Cohorts

Two study cohorts were used to evaluate the potential of AE-derived biomarkers. Cohort A was the phase I portion [37] with 32 patients from an ongoing phase I/II immunotherapy trial (NCT02638090) that evaluated the combination of HDAC inhibitor, vorinostat, and immunotherapy, pembrolizumab, in late-stage NSCLC. The cohort had 12 patients with PD, 18 patients with DC (three with PR and 15 with SD), and two patients not assessed. The distribution of DOT ranged from 0.37 to 24.4 months with a median of 2.85 months. The median survival time was 3.23 months for PFS and 7.37 months for OS. Cohort B was the dose escalation portion [38] with 50 patients from another ongoing phase I/IB immunotherapy trial (NCT02403193) that investigated the effect of Taminadenant (PBF509/NIR178), an Adenosine 2A Receptor antagonist, in patients with advanced NSCLC. Cohort B had 20 patients with PD, 25 patients with DC (2 CRs, 2 PRs, and 21 SDs), and five patients not assessed. The DOT distribution ranged from 0.07 to 70.7 months with a median of 1.87 months. The median survival time was 3.58 months for PFS and 8.63 months for OS. Both cohorts had available data of AE, response, and survival outcomes for analysis. Here, we investigated if the early AE-derived biomarkers could predict clinical outcomes in both cohorts.

## 3. Results

### 3.1. Cohort A

#### 3.1.1. Analysis of Overall AE Level

Survival outcomes: patients experienced with early low-grade overall TrAE (n = 22) were associated with longer PFS and OS compared to patients without the experiences (n = 10) (Figure 2A,B: median PFS: 5.7 vs. 1.7 months with *p* = 0.002; median OS: 10.4 vs. 3.7 months with *p* = 0.007). A higher frequency and longer duration of early low-grade overall TrAE were also associated with improved PFS (HR < 1 with *p* < 0.05 in Figure 3A). In contrast, patients experienced with early high-grade overall AE (n = 9) had shorter PFS and OS compared to the ones without the experiences (n = 23) (Appendix A: median PFS: 1.5 vs. 4.6 months with *p* = 0.0003; median OS: 2.7 vs. 8.8 months with *p* = 0.0002). Similarly, patients with higher frequency and longer duration of high-grade overall AEs had shorter PFS and OS (HR > 1 with *p* < 0.05 in Figure 3A). Early high-grade overall TrAE was also associated with poorer OS (HR > 1 with *p* < 0.05 in Figure 3A).

Treatment response: early low-grade overall TrAE had higher frequency and longer duration in patients with DC (n = 18) compared to patients with PD (n = 12) (mean duration: 26.7 vs. 9.8 days with *p* = 0.03; mean frequency: 1.6 vs. 0.6 AE events with *p* = 0.03 in Figure 2C and Figure 3B). On the other hand, patients with PD showed more experience or longer duration of early high-grade overall TrAE (*p* < 0.05; Figure 3B and Appendix A). Early high-grade overall TrAE was also correlated with shorter treatment duration (Appendix A).

**Figure 2 cancers-15-02521-f002:**
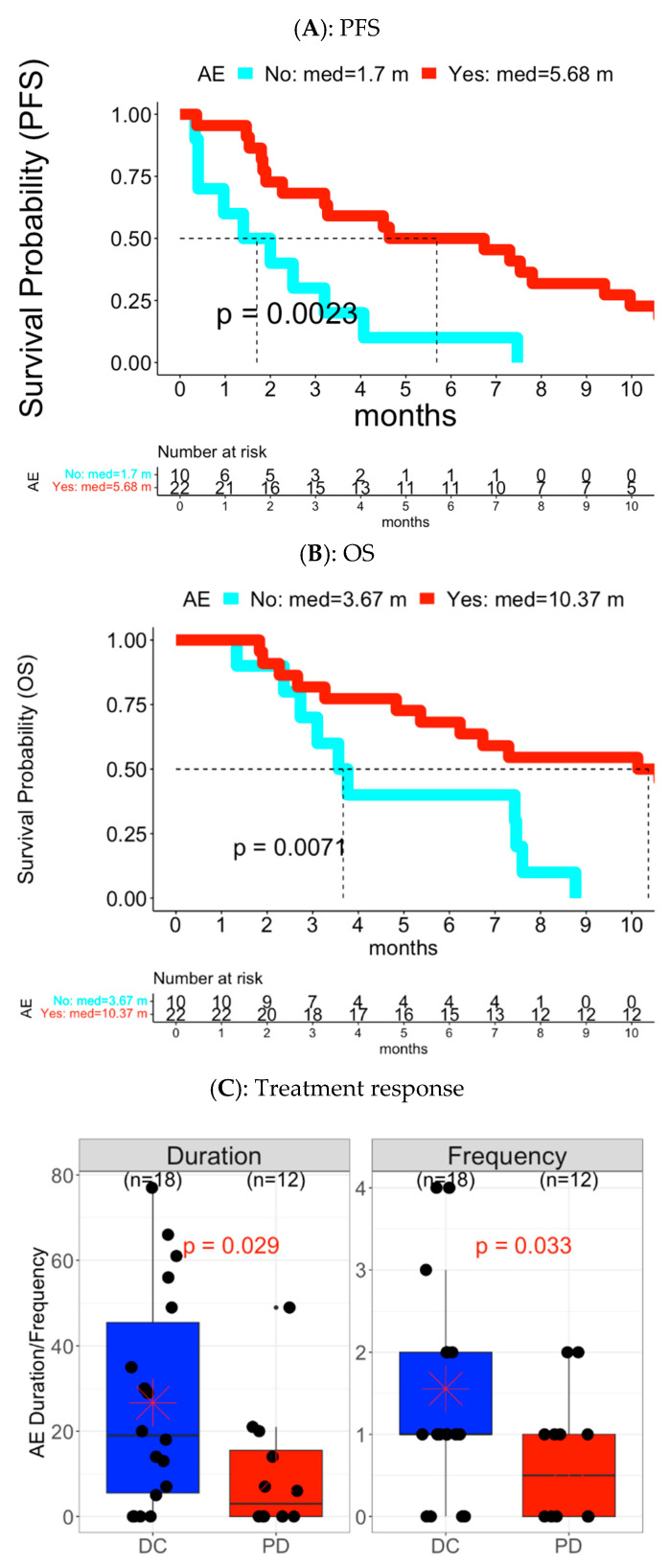
Early overall low-grade TrAE associated with PFS, OS, and treatment response in Cohort A. (**A**) PFS and (**B**) OS: The Y-axis was the survival probability while the X-axis was the time domain with month as unit. Red color curve represented the subgroup with experience of early AEs while the cyan color curve was the subgroup without experience of early AEs. The vertical dotted line was median survival time for each AE subgroup. The reported p value was based on log-rank test. The table below the survival curves was frequency of patients at risk at each time point for each AE subgroup. (**C**) Treatment response: Left panel was for distribution of AE duration while right panel was for distribution of AE frequency. Each panel had two boxplots: blue color for DC and red color for PD.

**Figure 3 cancers-15-02521-f003:**
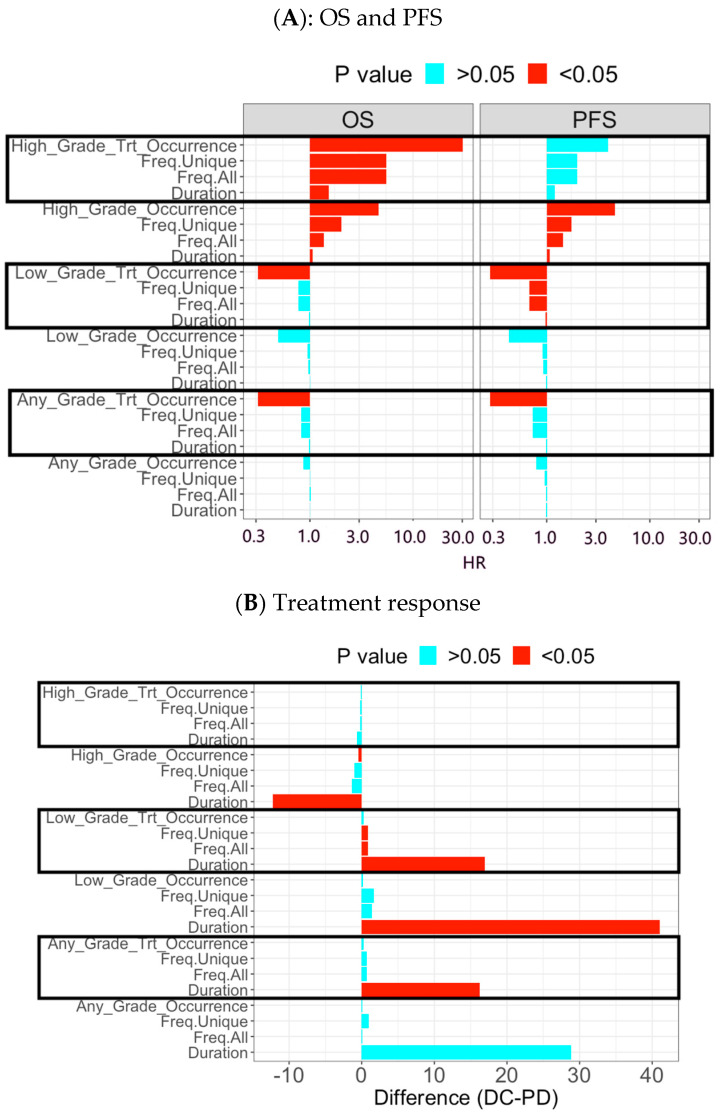
Summary plot to evaluate association of early AE-derived biomarkers in overall AE level with clinical outcomes in Cohort A.Y-axis represented the 24 AE-derived biomarkers which were grouped in six categories (high-grade, treatment related high-grade, low-grade, treatment related low-grade, any-grade, and treatment related any-grade). Each category included four measurement types of AE (occurrence, frequency of unique AEs, frequency of all AEs, and total duration). (**A**) OS and PFS: the X-axis was the effect size for hazard ratio (HR) and stratified by OS and PFS. HR > 1 indicates association with poorer survival outcome while HR < 1 implies positive association with survival outcome. Effect size with *p* < 0.05 was displayed in red color. For example, the treatment-related low-grade AE category had all the four measurement types of AE associated with improved PFS (HR < 1) and the occurrence event associated with improved OS. In contrast, the high-grade AE category had all the four measurement types of AE associated with poorer survival (OS and PFS; HR > 1); (**B**) treatment response: the X-axis was the effect size for mean difference of AE-derived biomarker between DC and PD. DC-PD > 0 indicates higher occurrence, frequency, duration in DC group while DC-PD < 1 implies the opposite direction. Effect size with *p* < 0.05 was displayed in red color. For example, the treatment-related low-grade AE category had the AE frequency and duration associated with DC (DC-PD > 0).

#### 3.1.2. Analysis of Toxicity Category Level and Individual AE Level

Survival outcome (Appendix A): there were three toxicity categories associated with improved survival outcomes. For PFS, endocrine disorders had early low-grade AE and TrAE associated with better PFS. Hypothyroidism, a member of endocrine disorders, was a Pembrolizumab immune-related adverse event (irAE), and shared the same association (median PFS: 13.4 months (with AE event; n = 3) vs. 3.2 months (without AE event) in Figure 4A). Eye disorders also had early low-grade AE associated with better PFS. For OS, low-grade gastrointestinal disorders and its member, diarrhea, were associated with better OS (median OS for diarrhea: 20.7 months (with AE event) vs. 6.73 months (without AE event) in Figure 4A). In contrast, there were eight toxicity categories and 15 individual AEs associated with poorer survival outcomes. For PFS, six toxicity categories and six individual AEs had high-grade or any-grade AE associated with poorer PFS: gastrointestinal disorders (vomiting), general disorders and administration site conditions, infections and infestations (lung infection), musculoskeletal and connective tissue disorders (back pain), “respiratory, thoracic and mediastinal disorders” (atelectasis and dyspnea), and vascular disorders (thromboembolic event). Psychiatric disorders (confusion) and “respiratory, thoracic and mediastinal disorders” (cough, pleural effusion, and pneumothorax) had low-grade AE associated with poorer PFS. For OS, TrAEs associated with poorer OS included high-grade AE in gastrointestinal disorders, and two members, diarrhea and vomiting, and low-grade AE in alanine aminotransferase increased. In addition, there were four toxicity categories and two individual AEs had high-grade AE associated with poorer OS: blood and lymphatic system disorders (anemia), general disorders and administration site conditions, infections and infestations, and “respiratory, thoracic and mediastinal disorders” (aspiration). Psychiatric disorders also had low-grade AE associated with poorer OS.

Treatment response (Appendix A): one toxicity category, general disorders and administration site conditions, with two associated AEs, fatigue and pain, had low-grade or any-grade AE associated with DC (Figure 4B). Another AE, platelet count decreased (n = 5; from investigations toxicity category), had low-grade Vorinostat-related TrAE associated with DC (Figure 4B).

Duration of treatment (Appendix A): there were five toxicity categories and six individual AEs with low-grade and any-grade events associated with longer treatment duration (Toxicity category: endocrine disorders, eye disorders, and skin and subcutaneous tissue disorders; individual AE: hypothyroidism, dry mouth, vomiting, dizziness, headache, and dry skin). Early low-grade TrAEs included hypothyroidism and vomiting.

### 3.2. Cohort B

#### 3.2.1. Analysis of Overall AE level

Patients experiencing early low-grade overall AE yielded better PFS and OS (Figure 5A,B). Moreover, patients with DC had a longer duration of early low-grade overall AE compared to patients with PD development (Figure 5C and Figure 6B). On the other hand, early high-grade overall AE was associated with poor PFS and OS (Figure 6A), including duration for both PFS and OS, and occurrence and frequency of unique AEs for OS.

#### 3.2.2. Analysis of Toxicity Category Level and Individual AE Level

Survival outcome (Appendix A): there was one toxicity category, gastrointestinal disorders, and the related AE, nausea, had early low-grade AE associated with improved PFS or OS (gastrointestinal disorders: median PFS = 3.97 months (with AE event) vs. 2.77 months (without AE event); median OS = 9.9 months (with AE event) vs. 4.43 months (without AE event) in Figure 7A). In contrast, six toxicity categories and the related 15 individual AEs were associated with poorer survival outcomes. The associated high-grade AEs included gastrointestinal disorders with two associated individual AEs, abdominal pain and dysphagia, general disorders and administration site conditions with two associated individual AEs, death NOS and fatigue, and “respiratory, thoracic and mediastinal disorders” with one associated AE, pleural effusion. The associated low-grade AEs included metabolism and nutrition disorders with two associated individual AEs, anorexia and hypoalbuminemia, psychiatric disorders with one associated individual AE, depression, “respiratory, thoracic and mediastinal disorders” with two associated individual AEs, cough and dyspnea, vascular disorders with one associated AE, hypertension, dry mouth and dysphagia (from gastrointestinal disorders), pain (from general disorders and administration site conditions), and back pain (from musculoskeletal and connective tissue disorders).

Treatment response (Appendix A): one toxicity category, gastrointestinal disorders, and the associated individual AE, nausea, had early low-grade AE and TrAE associated with DC (Figure 7B and Appendix A).

Duration of treatment (Appendix A): there were two toxicity categories and four individual AEs with low-grade or any-grade AE events associated with longer treatment duration (Toxicity category: skin and subcutaneous tissue disorders and investigations; individual AE: anemia, constipation, flatulence, rash maculo-papular).

## 4. Discussion

### 4.1. Informative Predictive Biomarkers

The study in both cohorts showed that early overall AE and TrAE were associated with clinical outcomes, indicating their potential predictive power. Patients experienced with low-grade overall AE or TrAE at early time point (at or before day 30 after on-treatment) had a better PFS, OS, or treatment response. In contrast, patients with early development of higher-grade overall AEs tended to have poorer PFS, OS, and with PD.

Specifically, Cohort A (treated with pembrolizumab and vorinostat) showed early low-grade TrAEs were associated with longer PFS or OS in overall AE level, in toxicity category level (endocrine disorders), and individual AE (hypothyroidism, an associated member of endocrine disorders). The median survival time was more than double in patients with AE experience compared to the ones without AE event experience (Figure 2A and Figure 4A). Hypothyroidism was shown to be a positive biomarker for clinical outcomes in cancer [39,40,41,42], including pembrolizumab treated patients with NSCLC [41]. Our result supports the evidence in early AE event content. For treatment response, platelet count decreased had Vorinostat treatment related low-grade AE associated with DC. Thrombocytopenia (platelet count decreased) was reported to be associated with improved survival in patients treated with immunotherapy [43]. In this cohort, the early AE event occurred in five SD patients with a duration ranging from 3 to 11 days, but not in PD patients. In terms of association with longer treatment duration, the relevant early low-grade TrAEs included hypothyroidism (n = 3) and vomiting (n = 3). Patients with hypothyroidism had treatments which lasted 187–483 days, while two patients with vomiting were on treatment for 226 and 733 days, respectively. In addition to early low-grade TrAEs, early low-grade AEs were also positively associated with clinical outcomes, including eye disorder event with better PFS, gastrointestinal disorders and its member, diarrhea, with better OS, fatigue and pain with DC, and dry mouth, dizziness, headache, and dry skin with longer treatment duration.

In contrast to AEs with improved clinical outcomes, there were different early AEs associated with negative effects. Specifically, patients with early high-grade overall TrAEs had shorter PFS and were associated with PD. Individual TrAEs correlated with worse OS included high-grade AE in gastrointestinal disorders with two members, diarrhea and vomiting, and low-grade AE in alanine aminotransferase increased (irAE) which was considered as a circulating biomarker of liver injury [44,45]. In addition, other early AEs with negative clinical outcomes included high-grade AEs in overall AE, six toxicity categories, and six individual AEs with poorer PFS, in overall AE, four toxicity categories, and two individual AEs with poorer OS, and low-grade AEs in two toxicity categories and four individual AEs with poorer PFS or OS.

We further analyzed the association in Cohort A between low- versus high-grade overall AE and found ten patients without experiences of early low-grade overall TrAE and six patients (60%) with early high-grade overall AE, compared to 22 patients experiencing early low-grade overall TrAE with only six patients (14%) with early high-grade overall AE, including one patient with early high-grade overall TrAE (*p* = 0.12 by Fisher exact test). The insignificant result suggests the early low-grade overall TrAE maintains unique predictive value even under mild association with high-grade overall AE.

In Cohort B (treated with Taminadenant), early low-grade TrAEs associated with positive clinical outcomes were gastrointestinal disorders and the member, nausea. Patients with DC (n = 25) tended to have longer AE duration compared to PD patients (n = 20). In addition, early low-grade overall AE yielded better OS and PFS and was associated with DC. Other low-grade AEs associated with better clinical outcomes included gastrointestinal disorders and nausea with improved PFS, and anemia, constipation, flatulence, rash maculo-papular with longer treatment duration. On the other hand, early AEs associated with negative survival outcomes included high-grade overall AE, three toxicity categories and five related individual AEs in high-grade AE level, and four toxicity categories and ten individual AEs in low-grade AE level.

Overall, the study showed evidence of AE biomarker utility in predicting positive and negative clinical outcomes. It could be TrAEs or a combination of treatment-related and not related AEs. Grade level played a key role in determining the direction of clinical outcomes with low-grade leaning to a positive effect and high-grade to a negative impact. Significant early AEs were identified from overall AEs, toxicity category AEs, down to individual AEs, providing comprehensive view of AE results to improve interpretation.

### 4.2. Unique AE Metrics and Full Spectrum of AE Data Analysis

By recognizing the underutilization of AE data due to the complexity of multi-parameters embedded in the data, this study took a novel approach to integrate various AE parameters and derived a set of innovative AE metrics. Specifically, a series of AE-derived biomarkers were developed based on severity level (grade) and treatment relatedness. Each of these biomarkers was further defined based on AE occurrence, frequency, and duration. By taking full consideration of the AE parameters, the AE-derived biomarkers present unique AE profiles for safety and efficacy analysis. The application to both cohorts demonstrated its feasibility by identifying clinical-associated AE biomarkers, such as early low-grade overall AEs and TrAEs with positive survival outcomes in cohort A. Subsequent analysis in toxicity category and individual AE led to discover endocrine disorders and the associated member, hypothyroidism, with improved PFS and longer treatment duration.

Moreover, global discovery analysis included survival outcomes, treatment response, and treatment duration for each AE-derived biomarker (e.g., Figure 2). The summary plot presents the effect size (e.g., HR) for all AE-derived biomarkers and colorfully highlights the significant ones (*p* < 0.05) (e.g., Figure 3). Given the large number of AEs, our unique algorithm generated concise tables (e.g., Appendix A) to efficiently summarize significant AEs and to gain insightful understanding of AE’s clinical impact. To our best knowledge, most AE reports in literature were either descriptive statistics covering all AEs to overwhelm readers or piecemeal results in contrast to our inclusive report. Furthermore, the AE-derived biomarkers were easy for calculation with straightforward interpretation, and therefore could be implemented directly in clinical practice.

## 5. Limitations and Future Directions

This study evaluated interim cohorts from two ongoing immunotherapy trials in NSCLC. Once both trials are completed, the new AE data with large sample size could be naturally served as validation sets to confirm if early AE-derived biomarkers could predict clinical outcomes. Additional AE data from other immunotherapy trials are also needed to make the results more generalizable. Another further direction is to evaluate AE in clinical trials using chemotherapy and targeted therapy. It would provide an opportunity to assess whether different therapy could yield distinctive AE biomarkers.

Moreover, the analysis generated multiple meaningful hypotheses worth future research: (1) the value of non-TrAEs. Various TrAEs were shown to have an association with clinical outcomes. Meanwhile, some AEs with combinations of treatment related and non-related events were also correlated with clinical outcomes (e.g., overall low-grade AE and nausea with improved PFS or OS). The results indicated the potential predictive and prognostic value of non-TrAEs. The separation of TrAE and non-TrAE events could help the assessment, but the process is not straightforward. The same AE could be a treatment related event or non-treatment related event within a patient (e.g., weight-loss in Patient A and fatigue in Patient B; Figure 1) or between patients. A thoughtful strategy is needed to dissect the two types of AE event; (2) sensitivity analysis of early AE event. The study used a length of 30 days to define early AE event and identified a series of AEs associated with clinical outcomes. However, it remains unclear whether the significant results would hold for different lengths. One potential approach is sensitivity analysis to help assess robustness of the significant AE biomarkers by evaluating various lengths (e.g., 2 weeks up to 6 weeks) and measuring variation of results for early AE event; (3) network analysis of clinically associated AEs: while significant AEs were identified, it would be more informative to evaluate their structural relations in terms of clinical relevance. Network analysis could be a useful utility to offer the potential for insight into AE relationship to inform clinical practice, similar to pathway analysis in bioinformatics; and (4) high correlation of AE-derived biomarkers. Correlation analysis showed moderate to high correlation between any-grade and low-grade overall AEs, but a weak negative correlation between high-grade and low-grade or any-grade overall AEs in frequency and duration. The utilization of machine learning methods could help to address the collinearity and multiple testing issues by data reduction techniques (e.g., principal component analysis and multi-dimensional scaling).

## 6. Conclusions

In precision medicine era, this study presents a paradigm-shifted approach for AE analysis from classic descriptive summary into modern informative statistics. It derives a series of innovative AE biomarkers by integrating multiple AE components from grade, treatment relatedness, occurrence, frequency, to duration. Across-the-board AE analysis generates opportunities for global discovery of predictive early AE-derived biomarkers from overall AEs, toxicity category AEs, to individual AEs. A proof of concept in two lung cancer studies demonstrates potential clinical utility of applying this new framework for identifying predictive AE biomarkers for treatment response and survival outcomes, which cannot be identified using standard paradigm. Moreover, the methodology modernizes AE data analysis by helping clinicians discover novel AE biomarkers to predict clinical outcomes and facilitate the generation of vast clinically meaningful research hypotheses in a new AE content to fulfill the demands of precision medicine.

## Figures and Tables

**Figure 1 cancers-15-02521-f001:**
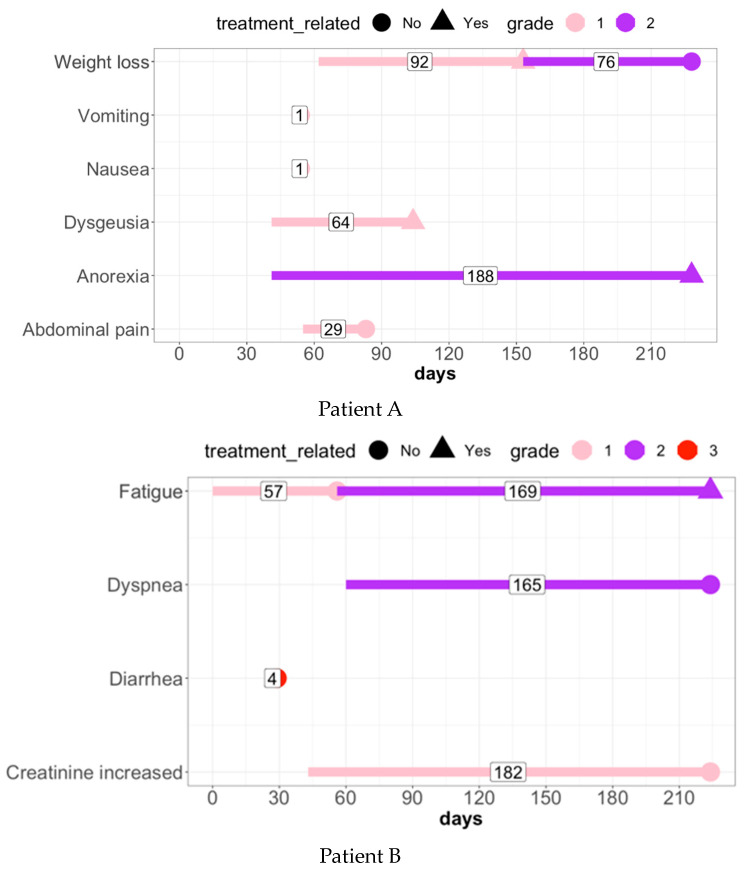
Complexity of AE data: Patient A had six unique AEs: weight loss, vomiting, nausea, dysgeusia, anorexia, and abdominal pain. Among them, weight lost was a repeated event from grade 1 advanced to grade 2. Treatment related AE included weight loss, dysgeusia, and anorexia. The worst grade method showed only two grade 2 AEs and four grade 1 AEs and was unbale to analyze (1) variation of duration (e.g., one day only for grade 1 vomiting and nausea versus 64 days for grade 1 dysgeusia; (2) recurrence event (e.g., weight loss); (3) change of grade and treatment relatedness over time (e.g., weight loss). Patient B had four unique AEs: fatigue, creatinine increased, dyspnea, and diarrhea. Fatigue was a repeated event from a non-treatment related grade 1 (57 days) advanced to treatment related grade 2 (169 days). Creatinine increased was a non-treatment related grade 1 event for 182 days and dyspnea was also a non-treatment related grade 2 for 165 days. In contrast, diarrhea was a non-treatment related grade 3 event with only 4 days. The worst grade method yielded, one grade 1 AE, two grade 2 AEs, and one grade 3 AE. The worst grade approach fails to capture (1) duration (e.g., same grade with different duration): while creatinine increased event and the first fatigue event were listed as grade 1, creatinine increased event (182 days) lasted longer than the fatigue (57 days); (2) recurrence: fatigue was a repeated event; (3) change of grade and treatment relatedness: initial fatigue was grade 1 and not associated with treatment, but then evolved to grade 2 and became treatment related.

**Figure 4 cancers-15-02521-f004:**
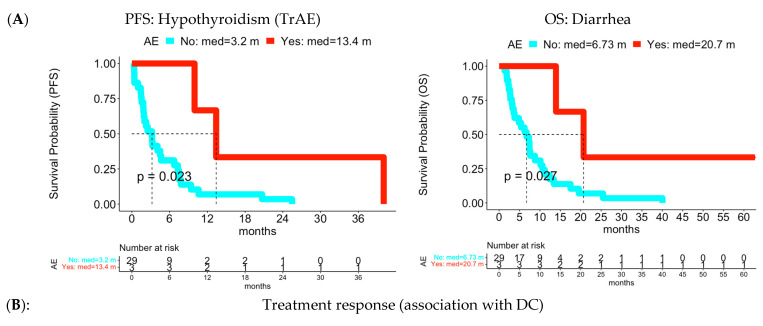
Individual low-grade early AEs/TrAEs associated with PFS, OS, and treatment response in Cohort A. (**A**) PFS for hypothyroidism and OS for diarrhea: The Y-axis was the survival probability while the X-axis was the time domain with month as unit. Red color curve represented the subgroup with experience of early AE while the cyan color curve was the subgroup without experience of early AE. The vertical dotted line was median survival time for each AE subgroups. The reported p value was based on log-rank test. The table below the survival curves was frequency of patients at risk at each time point for each AE subgroup. (**B**) Treatment response: Left panel was for distribution of low-grade fatigue AE, middle panel was for distribution of low-grade pain AE, and right panel was for distribution of low-grade trAE for platelet count decreased. Each panel had two boxplots: blue color for DC and red color for PD.

**Figure 5 cancers-15-02521-f005:**
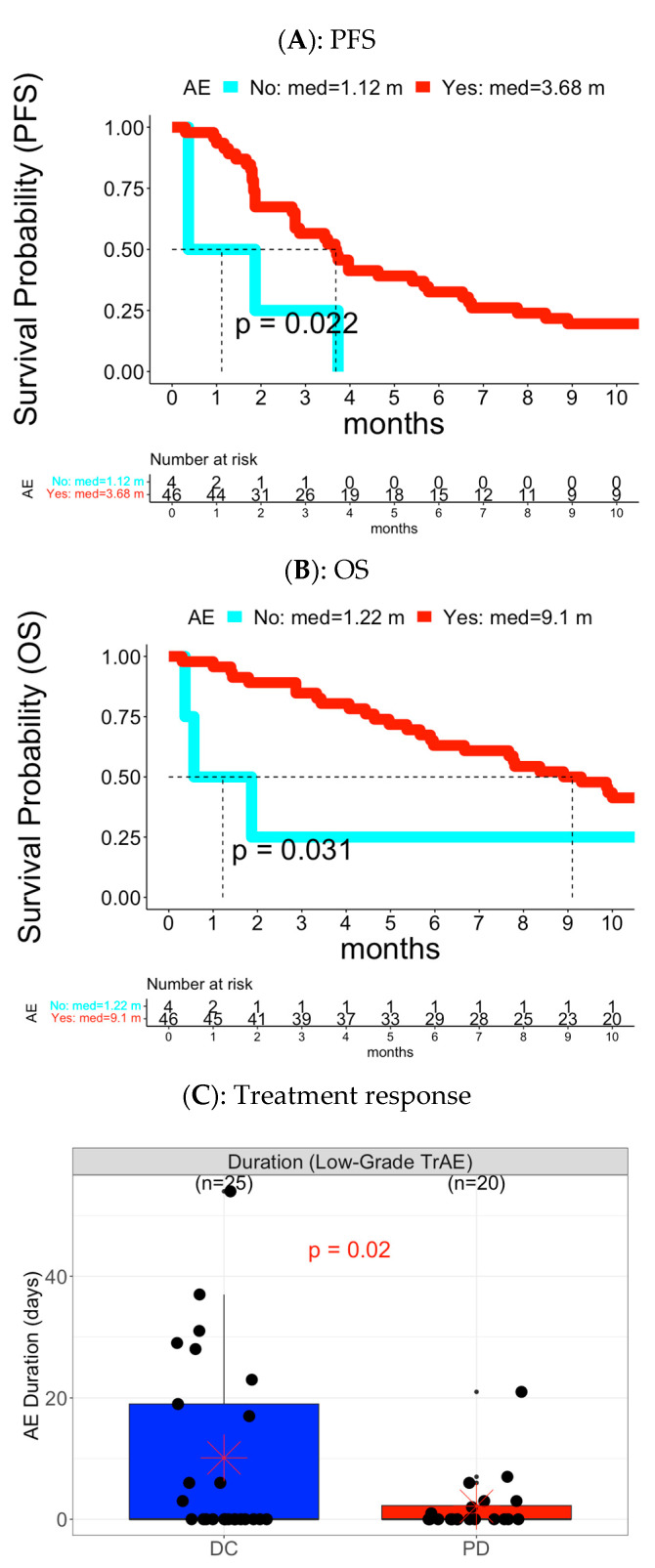
Early overall low-grade AEs associated with PFS, OS, and treatment response in Cohort B. (**A**) PFS and (**B**) OS: The Y-axis was the survival probability while the X-axis was the time domain with month as unit. Red color curve represented the subgroup with experience of early AEs while the cyan color curve was the subgroup without experience of early AEs. The vertical dotted line was median survival time for each AE subgroup. The reported p value was based on log-rank test. The table below the survival curves was frequency of patients at risk at each time point for each AE subgroup. (**C**) Treatment response: the panel was for distribution of AE duration with two boxplots, blue color for DC and red color for PD.

**Figure 6 cancers-15-02521-f006:**
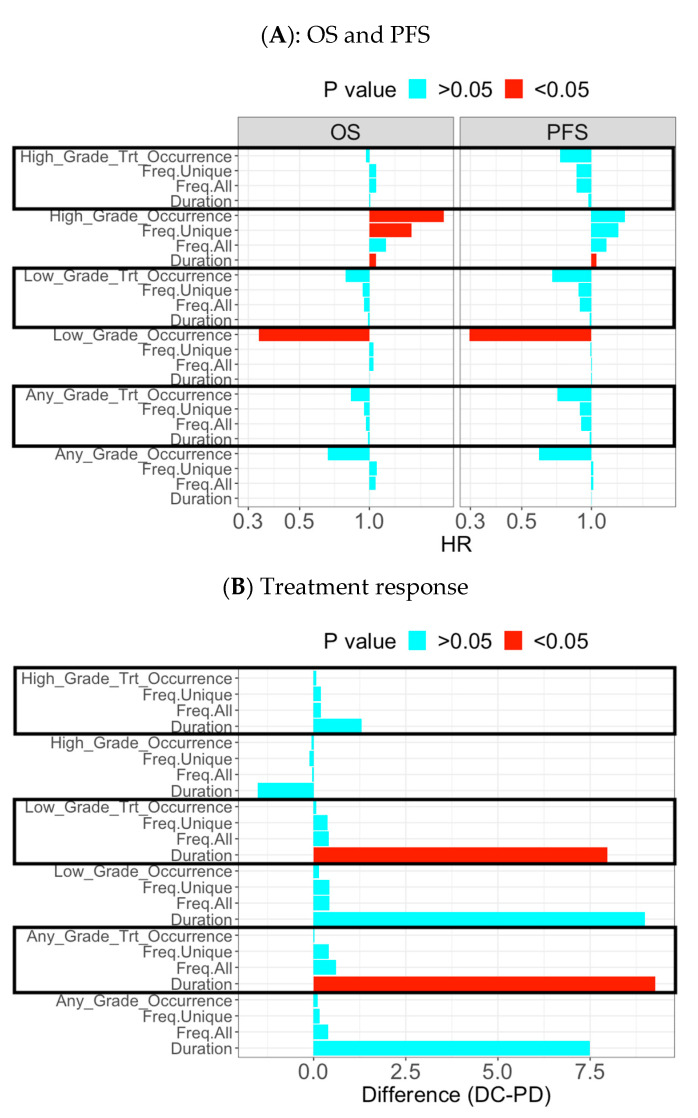
Summary plot to evaluate association of early AE-derived biomarkers in overall AE level with clinical outcomes in Cohort B: the Y-axis represented the 24 AE-derived biomarkers which were grouped in six categories (high-grade, treatment related high-grade, low-grade, treatment related low-grade, any-grade, and treatment related any-grade). Each category included four measurement types of AE (occurrence, frequency of unique AEs, frequency of all AEs, and total duration). (**A**) OS and PFS: the X-axis was the effect size for hazard ratio (HR) and stratified by OS and PFS. HR > 1 indicates association with poorer survival outcome while HR < 1 implies positive association with survival outcome. Effect size with *p* < 0.05 was displayed in red color. For example, low-grade AE category had the occurrence event associated with improved OS and PFS. In contrast, the high-grade AE category had occurrence, frequency of unique AEs, and duration associated with poorer OS (HR > 1). The AE duration was also negatively associated with PFS. (**B**) Treatment response: the X-axis was the effect size for mean difference of AE-derived biomarker between DC and PD. DC-PD > 0 indicates higher occurrence, frequency, duration in DC group while DC-PD < 1 implies the opposite direction. Effect size with *p* < 0.05 was displayed in red color. For example, the treatment-related low-grade AE category had the AE duration associated with DC (DC-PD > 0).

**Figure 7 cancers-15-02521-f007:**
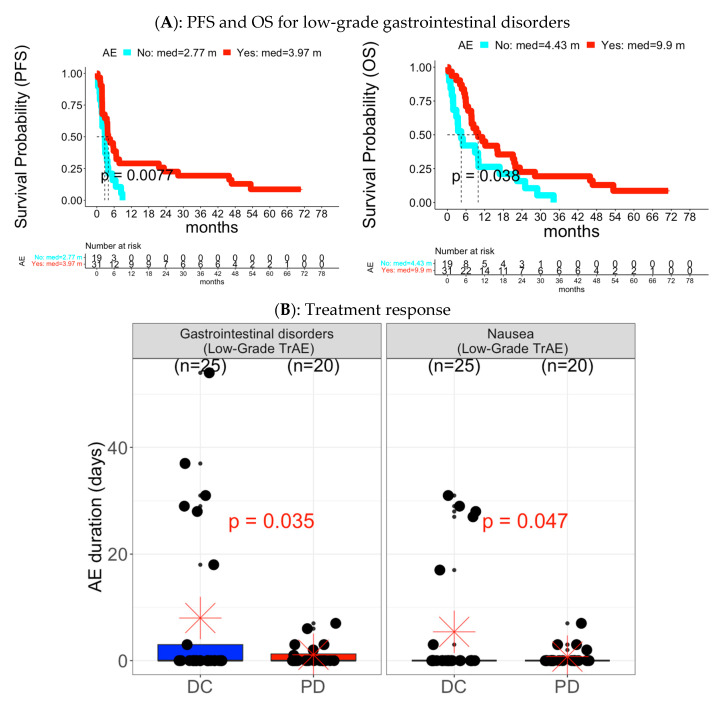
Individual low-grade AEs (Gastrointestinal disorders) associated with PFS, OS, and treatment response in Cohort B. (**A**) PFS and OS for low-grade gastrointestinal disorders: The Y-axis was the survival probability while the X-axis was the time domain with month as unit. Red color curve represented the subgroup with experience of early AE while the cyan color curve was the subgroup without experience of early AE. The vertical dotted line was median survival time for each AE subgroup. The reported p value was based on log-rank test. The table below the survival curves was frequency of patients at risk at each time point for each AE subgroup. (**B**) Treatment response: Left panel was for distribution of low-grade gastrointestinal disorders, and right panel was for distribution of low-grade nausea. Each panel had two boxplots: blue color for DC and red color for PD.

**Table 1 cancers-15-02521-t001:** AE-derived biomarkers.

		Measurement Type			
		Occurrence	Sum of All Unique AEs	Sum of All AEs	Sum of All AE Duration
Grade/ Treatment relatedness	Any-grade	x	x	x	x
	Treatment related any-grade	x	x	x	x
	Low-grade (1 or 2)	x	x	x	x
	Treatment related low-grade	x	x	x	x
	High-grade (3 or higher)	x	x	x	x
	Treatment related high-grade	x	x	x	x

Derivation of AE biomarkers started with combination of three types of grades and two types of treatment relatedness to form six categories (row-wise): any-grade, treatment related any-grade, low-grade, treatment related low-grade, high-grade, and treatment related high-grade. Each category included four measurement types of AE (column-wise: occurrence, frequency of unique AEs, frequency of all AEs, and total duration).

## Data Availability

AE data were deposited in Github: https://github.com/dungtsa/Adverse-Event-Data-Analysis, accessed on 18 April 2023.

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
