# Peer review of "Early Adverse Event Derived Biomarkers in Predicting Clinical Outcomes in Patients with Advanced Non-Small Cell Lung Cancer Treated with Immunotherapy"

_cancers, 2023, doi:10.3390/cancers15092521_

Round 1
Reviewer 1 Report
Dear Authors,
I think, the study is a little nice approach in perspective how to use AE data beyond the clinical-trial pharmacovigilance need.
Quality of presentation: The text body is really well written. In contrast, the figures are poor. Enough and correct, but you need to dig in. The authors could make it easier for the scientist how are interest in their work.
I think also, that the most valuable impact is the applicability and reconstruction of their method for own analysis. Due to this, I tried to give some guidance to improve their method and experience description for the interested audience.
So, my comments are more into direction of application, than scientific improvement (here the format is matching to motivation and message).
Author Response
Dear Authors,
I think, the study is a little nice approach in perspective how to use AE data beyond the clinical-trial pharmacovigilance need.
Quality of presentation: The text body is really well written. In contrast, the figures are poor. Enough and correct, but you need to dig in. The authors could make it easier for the scientist how are interest in their work.
I think also, that the most valuable impact is the applicability and reconstruction of their method for own analysis. Due to this, I tried to give some guidance to improve their method and experience description for the interested audience.
So, my comments are more into direction of application, than scientific improvement (here the format is matching to motivation and message).
'The authors should improve the tables and graphics in better way to make it easier or more intuitive to catch the information for readers.'
Response: We thank the reviewer’s constructive comment. We have added relevant description in figure and table to self-explain the information. For example, Table 1 is for derivation of AE biomarkers. We included the text to explain how the 24 AE biomarkers were developed. Similarly, Figure 3 and 6 are the summary plots. A new text was included to guide reader to interpretate the results.
Reviewer 2 Report
The manuscript reads well except for the same small parts of the subsections. I have highlighted them for the benefit of the readers.
1. Please briefly express the type of statistical analysis performed for the study in the methods section of the abstract.
2. Figure captions should not be part of the figure. please correct it. Also, minimize the text within the figures.
3. Please revise the statement "limitations have prevented their board application due to either limiting to single AE analysis, lack of full utilization of all AE parameters, difficulty of interpretation, or uncertainty of clinical relevance" by using alternative words to prevent repeating.
4. Please revise the following statement "majority of patients without the AE event will be dropped in the analysis. "
5. Please rephrase the the following statement "Limitation in-cludes dependency of utility weights which could yield different results, lack of estimation at a specific time point which becomes less predictive value, and inability in individual AE analysis. "
6. You're not expressing the study's hypothesis to the end of the introduction.
7. Please don't use background color for figures and tables.
Author Response
Reviewer 2
Comments and Suggestions for Authors
The manuscript reads well except for the same small parts of the subsections. I have highlighted them for the benefit of the readers.
Response: We thank the reviewer for valuing this study and appreciate the insightful comments. Below are our responses.
- Please briefly express the type of statistical analysis performed for the study in the methods section of the abstract.
Response: We have included the statistical methods in the abstract.
“Methods. We utilized a set of AE associated parameters (grade, treatment relatedness, occurrence, frequency, and duration) to derive 24 AE biomarkers. We further innovatively defined early AE biomarkers by landmark analysis at early time point to assess the predictive value. Statistical methods included Cox proportional hazards model for progression-free survival (PFS) and overall survival (OS), two-sample t-test for mean difference of AE frequency and duration between disease control (DC: complete response (CR) + partial response (PR) + stable disease (SD)) versus progressive disease (PD), and Pearson correlation analysis for relationship of AE frequency and duration versus treatment duration. Two study cohorts (Cohort A: vorinostat + pembrolizumab, and B: Taminadenant) from two immunotherapy trials in late-stage non-small cell lung cancer were used to test the potential predictiveness of AE-derived biomarkers. Data over 800 AEs were collected per standard operating procedure in a clinical trial using the Common Terminology Criteria for Adverse Events v5 (CTCAE). Clinical outcomes for statistical analysis included PFS, OS, and DC.”
- Figure captions should not be part of the figure. please correct it. Also, minimize the text within the figures.
Response: We have deleted caption, minimized text, and reformatted the figures to improve readability. We also took Reviewer 1’s advice to improve interpretability of figure and table by adding an independent text box below the figure and table with relevant description to self-explain the information (e.g., Table 1 and Figure 3). The layout will let (1) knowledgeable researcher directly jump to plot or table by skipping the text and (2) general reader use the text as guidance to interpretate the results.
- Please revise the statement "limitations have prevented their board application due to either limiting to single AE analysis, lack of full utilization of all AE parameters, difficulty of interpretation, or uncertainty of clinical relevance" by using alternative words to prevent repeating.
Response: We have revised the sentence to avoid duplication of same words.
“Their broad application was hindered due to either restriction to single AE analysis, less optimization of AE parameters, difficulty of interpretation, or concern of clinical relevance.”
- Please revise the following statement "majority of patients without the AE event will be dropped in the analysis. "
Response: We have revised the sentence for clarifcation.
“time-to-event analysis for AE onset could be biased because AE could occur only to a few patients (spareness) causing majority of patients without AE event excluded in the analysis.”
- Please rephrase the following statement "Limitation in-cludes dependency of utility weights which could yield different results, lack of estimation at a specific time point which becomes less predictive value, and inability in individual AE analysis. "
Response: We have revised the sentence to avoid duplication of same words.
“Concerns include dependency of utility weights which could influence results, lack of estimation at a specific time point which could diminish predictive value, and inability of individual AE analysis.”
- You're not expressing the study's hypothesis to the end of the introduction.
Response: We have added the hypothesis at the end of the introduction to highlight the study objective.
“By taking these considerations with the hypothesis of AE’s clinically predictive value, an integrative approach was developed by utilizing AE associated parameters and derived a set of innovative AE biomarkers. Global discovery analysis showed AEs as potential biomarkers in predicting clinical outcomes.”
Please don't use background color for figures and tables.
Response: We have re-created figures and tables without background color.